# Culturally Informed Australian Aboriginal and Torres Strait Islander Evaluations: A Scoping Review

**DOI:** 10.3390/ijerph20146437

**Published:** 2023-07-24

**Authors:** Kristina Vine, Tessa Benveniste, Shanthi Ramanathan, Jo Longman, Megan Williams, Alison Laycock, Veronica Matthews

**Affiliations:** 1University of Sydney, University Centre for Rural Health, Lismore, NSW 2480, Australia; jo.longman@sydney.edu.au (J.L.); alison.laycock@sydney.edu.au (A.L.); veronica.matthews@sydney.edu.au (V.M.); 2School of Health, Medical and Applied Sciences, CQUniversity, Adelaide, SA 5034, Australia; t.benveniste@cqu.edu.au; 3Hunter Medical Research Institute, Newcastle, NSW 2305, Australia; shanthi.ramanathan@hmri.org.au; 4College of Health, Medicine and Wellbeing, University of Newcastle, Newcastle, NSW 2300, Australia; 5School of Public Health, University of Technology Sydney, Sydney, NSW 2007, Australia; megan.williams@uts.edu.au

**Keywords:** evaluation, Aboriginal and Torres Strait Islander peoples, community engagement, community led, culturally informed

## Abstract

Rigorous and effective evaluations inform policy and service delivery and create evidence of program impacts and outcomes for the communities they are designed to support. Genuine engagement of communities is a key feature of effective evaluation, building trust and enhancing relevancy for communities and providing meaningful outcomes and culturally relevant findings. This applies to Indigenous peoples’ leadership and perspectives when undertaking evaluations on programs that involve Indigenous communities. This systematic scoping review sought to explore the characteristics of culturally informed evaluations and the extent of their application in Australia, including the use of specific evaluation tools and types of community engagement. Academic and grey literature were searched between 2003 and 2023, with 57 studies meeting the inclusion criteria. Over time, there was an increase in the number of culturally informed evaluations undertaken, predominantly in the health and wellbeing sector. Around a quarter used a tool specifically developed for Indigenous evaluations. Half of the publications included Indigenous authorship; however, most studies lacked detail on how evaluations engaged with communities. This review highlights the need for further development of evaluation tools and standardised reporting to allow for shared learnings and improvement in culturally safe evaluation practices for Aboriginal and Torres Strait Islander communities.

## 1. Introduction

Rigorous and effective evaluations are important for developing evidence to inform policy and improve services and program delivery. Done well, evaluations also form an evidence base of impact and outcomes on the individuals and communities that the program or service targets. But what does an evaluation ‘done well’ look like? What are the characteristics of a high-quality evaluation?

In the context of policies and programs that are designed to serve Indigenous communities, that question leads us to consider the socio-historical and cultural context and needs of the communities themselves. As Waapalaneexkweew and Dodge-Francis write, “*Evaluation should be a tool of transformation, improvement, and empowerment to solve chronic issues in society. Inclusion of Indigenous theories and methods, Tribal governments, and Indigenous people … needs to be at the front end of this process, not an afterthought.*” [1] (p. 27). Evaluations led by and embedded within Indigenous peoples’ contexts have the potential to help improve program and service responsiveness to communities’ needs and develop an evidence base to inform policy that will produce outcomes of value to Indigenous communities [2,3]. 

Evaluating programs also fulfil the need for accountability—to show what impacts have been made and if intended outcomes have been achieved. This aspect of accountability, or as Hudson [4] recommends, co-accountability, from policy makers, program funders and program providers is critical for securing ongoing funding and resources, as well as allowing for open and transparent feedback to participants and other stakeholders. Australia’s national Aboriginal and Torres Strait Islander health research institute, the Lowitja Institute, also acknowledges the importance of undertaking evaluation in transferring knowledges within and between Indigenous communities and informing strengths-based public discourse and policy [5]. 

Across various sectors, there is growing recognition of the importance of incorporating Indigenous ways of knowing, being and doing into all aspects of research and knowledge generation for services, programs and policies that relate to Indigenous peoples [6]. Indigenous research methodologies contrast with Western approaches in many fundamental ways, with First Nation holistic approaches centred around relationality, respect and reciprocity—acknowledging the importance of connection to and caring for land, culture and kin [7]. However, there remains limited information and guidance on embedding Indigenous methodologies into evaluation processes across culturally diverse Indigenous contexts [1,3,8,9]. 

The use of Indigenous methodologies in the context of evaluation is one of the components for supporting culturally safe evaluations—evaluations that are “*planned and implemented in a way that is safe, respectful and valuable for Aboriginal and Torres Strait Islander peoples who are involved and impacted by the stories that evaluations tell*” [5] (p. 1). Central also in striving to ensure cultural safety in engaging with the Indigenous communities, whom the evaluated programs and services are designed to serve, is being aware of the ‘who’ that are leading the approach and undertaking the evaluation. When non-Indigenous ‘outsiders’ determine the scope and methodologies of a study without Indigenous consultation and meaningful participation and leadership, there are a range of inherent risks posed. These can include further colonisation of Indigenous knowledge, data and storytelling through imposing Western worldviews; harm done where work is not undertaken appropriately or with sensitivity to local context; and deepening power imbalances that limit the potential of Indigenous sovereignty [3,8,10]. All of these potential risks can reduce the validity and reliability of an evaluation [3]. Efforts to safeguard against these risks focus on ensuring Indigenous people are included in evaluation leadership teams; good governance and ensuring accountability to Indigenous communities; and working within Indigenous frameworks and using Indigenous research methodologies throughout the evaluation processes. These safeguards require partnerships that are supported by respectful two-way approaches between Indigenous and non-Indigenous evaluators and non-Indigenous evaluators’ commitment to practices of critical self-reflection and accountability [8,10,11,12]. Historically, large formal program and policy evaluations have been undertaken by ‘outsiders’, i.e., external contractors or consultants. This can be at significant cost and requires long timeframes and the mutual commitment of evaluators and participants to develop trusting relationships and shared understandings. 

A preferable approach is through developing evaluation skills at a local community level can help provide community autonomy in evaluation processes. Local decision making on the when, who, what and how in program design, implementation and outcomes will more likely lead to valuable and relevant findings to make improvements to service delivery or to advocate for greater resourcing or policy change. Greater Indigenous ownership of the evaluation process also allows them to set their own local program agendas, develop evidence that is contextually based, have autonomy in the use of evaluation data and improve services and community outcomes [8,11].

Internationally, there has been a growth in the number of Indigenous-led, formally funded evaluations and an increasing trend of cross-cultural collaborations in evaluations (i.e., Indigenous and non-Indigenous partnering), as well as the development of culturally responsive evaluation resources and guiding principles [3,13,14]. Within Australia, the need for practice guidance for undertaking culturally competent evaluations in the context of Aboriginal and Torres Strait Islander values has long been identified [15,16]. However, over the past decade, there has been limited guidance and synthesis of findings and learnings from evaluations that have utilised resources informed by Aboriginal and Torres Strait Islander worldviews and experience or that have sought to centralise and be led by community voices and priorities [17]. From roundtable discussions on the role that evaluations can have on Indigenous policy in 2012, the Australian Productivity Commission noted that due to the lack of mandated and effective evaluations, there are significant gaps in the evidence base of the impacts of Australian-Government-funded policies and programs [16]. It was also acknowledged that the usual evaluation methodologies failed to include Aboriginal and Torres Strait Islander peoples’ perspectives and participation and that any evaluation should ensure technical and financial resourcing to necessitate Aboriginal and Torres Strait Islander people’s self-determination and empowerment [16]. These evidence gaps exist despite the introduction in 2003 of national ethical guidelines to follow when conducting Aboriginal and Torres Strait Islander research, including evaluations [18]. Since then, there has been positive progress in the development of new national evaluation resources [2,12,14,19]. One example is the development in 2020 of the Indigenous Evaluation Strategy, which aims to make Aboriginal and Torres Strait Islander perspectives, knowledge and priorities central to evaluation design and emphasizes the need for culturally appropriate evaluation methods and tools in guiding policy and practice [19].

With this positive shift occurring in Australia, this paper aims to contribute to the evidence and guidance for undertaking evaluations that are safe and responsive to Aboriginal and Torres Strait Islander cultures. This review of the literature focuses on the type of Indigenous evaluation frameworks, guidelines and tools used in the Australian context and how Aboriginal and Torres Strait Islander communities are engaged in the development and implementation of evaluations that pertain to their communities. 

This review is a collaboration of Indigenous (MW, VM) and non-Indigenous researchers (KV, TB, SR, JL, AL) who collectively have extensive experience working alongside Aboriginal and Torres Strait Islander communities and community-based health organizations. This authorship group is committed to finding better ways of undertaking culturally safe and relevant evaluations that foreground Indigenous ways of knowing being and doing and promote Indigenous leadership alongside conscious and respectful allyship.

For the Australian context, we use the term ‘Aboriginal and Torres Strait Islander’ throughout and ‘Indigenous’ and ‘First Nation’ when referring to any and all First Nation peoples internationally. 

## 2. Materials and Methods

This review systematically scoped the literature for studies describing evaluation guides, frameworks, tools, resources or processes that have been informed by, or applied by, First Nation communities and cultures. Refinement to literature within the Australian context and to only include Aboriginal and Torres Strait Islander peoples was applied during screening based on reviewer discussions and informed by the large volume of results, and the process followed the PRISMA extension for scoping reviews (PRISMA-Scr) [20]. 

### 2.1. Search Strategy 

In line with the PRISMA-Scr guidelines, an exploratory pilot search was conducted (KV) initially in Scopus and Google databases, as well as using the online citation-based literature mapping tool ResearchRabbit [21] to determine the appropriate timeframe for inclusions, which was set as January 2003–January 2023 based on pilot results, and to further refine the search terms based on retrieved titles and subject headings. 

A full search was then completed for all published peer-reviewed literature indexed in Scopus, CINAHL, Informit and OVID by Medline databases using subject headings and keywords relating to four concepts, as detailed in Table 1: First Nations peoples; evaluations; resources; and culturally informed. Grey literature was searched in Google using the same search strings, with the first 100 hits screened and restricted to pdf file types. Items published in English were grouped in EndNote 20 [22] and exported into the online tool Covidence [23] for the removal of duplicates. 

Literature was reviewed for inclusion through a two-step process using Covidence. Abstract and title screening was undertaken by a single reviewer (KV), followed by the assessment of full-text data against the following inclusion and exclusion criteria, which was double screened by two reviewers (KV, TB), with conflicts solved through collaborative discussion and reviewer consensus. Studies were included if they described the characteristics of an evaluation that was culturally informed, which this review determined as an evaluation undertaken in the Australian context that was informed by Aboriginal and/or Torres Straits Islander culture through community engagement or the application of an Indigenous-specific evaluation resource. Studies were excluded if they did not meet that inclusion criterion, i.e., studies did not look at an evaluation, the evaluation was not undertaken in Australia, the evaluation did not include mention of Aboriginal and/or Torres Strait Islander people in the study population and did not either describe any community engagement as part of the evaluation or did not identify the use of an Indigenous-specific evaluation resource. Study protocols and studies describing evaluation plans prior to implementation were also excluded. 

**Table 1 ijerph-20-06437-t001:** Search strategy combining four key concepts used for Scopus, CINAHL, Informit, OVID by Medline and grey literature search via Google.

Concept 1—First Nations peoples
“Oceanic ancestry group*” OR Indigenous OR Aborigin* OR “Torres Strait Island*” OR “australian race” OR australoid OR “First Nation*” OR Metis* OR Inuit* OR Native* OR Maori OR “American Indian*” OR “Alaska Native*” OR “Native Hawaiian”
AND
Concept 2—Evaluation
evaluation OR monitoring OR impact OR “program evaluation” OR “developmental evaluation” OR “process evaluation” OR “implementation evaluation” OR “impact evaluation” OR “economic evaluation” OR “participatory evaluation” OR “empowerment evaluation”
AND
Concept 3—Resources
Guideline* OR framework* OR resource* OR tool* OR “program tool*” OR “program resource*” OR process*
AND
Concept 4—Culturally informed
“community led” OR “community engagement” OR “community participation” OR co-design OR “community driven” OR self-governance OR self-determination OR “Community network*” OR “indigenous knowledge*” OR “cultural* appropriate” OR “cultural* informed” OR “cultural* responsive*” OR “cultural* safe*” OR “participatory action research”

* Indicates where truncation was used to broaden the search of the root word.

### 2.2. Study Inclusion and Information Extraction

Included studies were extracted collectively (KV, TB, VM) using a matrix developed in Microsoft Excel to identify the characteristics of the evaluations, including modes of Indigenous engagement, leadership and use or development of any evaluation resources. All included literature for this review is listed in Appendix A, Table A1. Data were then categorized into thematic groups to summarise the main findings, Table 2, Table 3 and Table 4. 

## 3. Results

We identified 57 studies that met the criteria for this scoping review (Figure 1). There was a trend of increased publications over time across the two decades, with no inclusions from 2003 to the highest number of seven in each of the years 2020, 2021 and 2022 (Figure 2). There were also two inclusions in the first month of 2023, which are not depicted in Figure 2 as this represents only a single month of data. 

### 3.1. Study Characteristics

Overall, the majority of studies related to health and wellbeing evaluations (72%), predominantly for health promotion programs and health service delivery (Table 2). There was a far smaller representation from education (7%) and community services (7%) with smaller numbers of studies in the fields of other government sector service delivery such as justice and transport (5%) and land and sea management (5%). Two papers (4%) discussed the development of new evaluation resources through broad cross-sectorial engagements.

**Table 2 ijerph-20-06437-t002:** Sectors of included evaluation studies.

Sector of Evaluation Study	Number	Summary of Sector Sub-Topics	Relevant Studies
Health and Wellbeing	41 (72%)	Health promotion, health service delivery, health education, social and emotional well-being programs and specific health condition topics such as mental health or maternal and child healthcare	(Bennie 2021 [24]; Black 2019 [25]; Blignault 2016 [26]; Campbell 2018 [27]; Copley 2021 [28]; Crooks 2022 [29]; Doyle 2016 [30]; Durey 2016 [31]; Entwistle 2009 [32]; Freene 2021 [33]; Guenther 2022 [34]; Gwynne 2022 [35]; Haynes 2019 [36]; Hill 2020 [37]; Janca 2015 [38]; Jongen 2020 [39]; Kelaher 2018 [2]; Kildea 2009 [40]; Kildea 2012 [41]; Lopes 2012 [42]; Macniven 2023 [43]; Marchetti 2022 [44]; Mitchell 2021 [45]; Nagel 2020 [46]; O’Donnell 2020 [47]; Queensland Health 2020 [48]; Reilly 2011 [49]; Rowley 2015 [50]; Santhanam 2006 [51]; Saunders 2022 [52]; Stajic 2019 [53]; Tane 2022 [54]; Taylor 2012 [55]; Tsey 2004 [56]; Tsey 2010 [57]; Victorian Government 2019 [58]; West 2021 [59]; Williams 2018 [60]; Williams 2023 [17]; Xu 2018 [61]; Young 2019 [62])
Education	4 (7%)	Cultural safety in education and delivery of education programs	(Howard 2017 [63]; Kowanko 2009 [64]; Mills 2022 [65]; Purdie 2004 [66])
Community services	4 (7%)	Family services, out of home care services and non-profit organisations	(Haviland 2012 [67]; Lawton 2020 [68]; Rogers 2017 [69]; Rogers 2018 [70])
Other Government sector services	3 (5%)	Transport, justice and local delivery of employment and social services	(Cullen, 2016 [71]; Hurley 2004 [72]; Young, 2013 [73])
Land and Sea Management	3 (5%)	Natural resource management environmental programs	(Austin 2017 [74]; Robinson 2021 [9]; Sithole 2007 [75])
Cross-sector evaluations	2 (4%)	National and whole of government approaches	(Gollan, 2021 [12]; Productivity Commission, 2020 [19])

### 3.2. Use of Applied, Adapted or Developed Evaluation Resources

For this review, we collectively term any evaluation tools, frameworks or guidelines as ‘resources’. As seen in Figure 2, the results show a trend towards a greater number of relevant evaluations being published over time, particularly over the last three years. There is great diversity in the reported resource types utilised. Overall, only half (50%) of included literature used a specific evaluation resource, with other studies describing their methodology without reference to any evaluation tool, framework or guideline but instead noted a mode of community engagement (Table 3). Among those that did reference an evaluation resource, 15 studies (26%) utilised a resource that was developed specifically for an Aboriginal and Torres Strait Islander context, referred to here as ‘Indigenous specific resources’. Some of the other reported resources were themselves clinical tools for use with individuals, e.g., the Kessler 10 and EuroQoL, that were used as part of an evaluation.

Indigenous-specific resources are the largest category referenced in the studies, all published within the last eight years. This is an encouraging finding, suggesting there is a recent trend towards utilization of culturally relevant tools, guidelines and framework (26%) in comparison to using mainstream-only resources that have not been adapted in any way (12%) or modifying mainstream tools (11%) to fit the cultural context.

**Table 3 ijerph-20-06437-t003:** Evaluation resources applied, adapted or developed in included studies.

Evaluation Resources ^1^	Number	Reported Resources Used	Relevant Studies (by Year of Publication)
**Developed specifically for Indigenous Evaluations**			
Framework	9 (16%)	Nga-bin-ya Evaluation Framework; Evaluation framework developed as part of evaluation; the Indigenous Land and Sea Management (ILSM) Outcomes Framework; the AES First Nations cultural safety framework; the Nargneit Birrang framework service design, implementation and evaluation; Evaluation framework to Improve Aboriginal and Torres Strait Islander Health, RISE Framework	(Bennie 2021 [24]; Robinson, 2021 [9]; Gollan 2021 [12]; Queensland Health 2020 [48]; Lawton 2020 [68]; Victorian Government 2019 [58]; Kelaher 2018 [2]; Williams 2018 [60]; Howard 2017 [63])
Guideline	1 (2%)	Indigenous Evaluation Strategy	(Productivity Commission 2020 [19])
Tool	5 (9%)	Growth and Empowerment Measure (GEM) Tool; non-specific evaluation tool developed as part of evaluation such as checklist or data collection tool; the Cultural Identity Intervention Systematic Review Proforma (CIR) tool; Australian Outcome Measure for Indigenous Clients (ATOMIC); Here and Now Aboriginal Assessment (HANAA)	(Tane, 2022 [54]; Copley 2021 [28]; Hill, 2020 [37]; Rogers 2018 [70]; Blignault 2016 [26]; Doyle, 2016 [30]; Janca 2015 [38])
*Total*	*15 (26%)*		
**Adapted for Indigenous context**			
Framework	2 (4%)	The Rambaldini Model; modified framework based on the Community Capacity Index	(Gwynne, 2022 [35]; Entwistle 2009 [32])
Guideline	0	-	-
Tool	4 (7%)	Revised Cultural Capability Measurement Tool (CCMT); adapted Tri-Ethnic Research Centre’s Community Readiness Tool (CRT); Outcome Measures App developed based on adaptations and translation of tools Kessler 10, Patient Health Questionnaire 9 and EuroQoL; Gawugaa-gii-mara (head, heart, hands)	(Williams, 2023 [17]; Mills, 2022 [65]; West, 2021 [59]; Nagel 2020 [46])
*Total*	*6 (11%)*		
**Mainstream resource applied to Indigenous context (no adaption)**			
Framework	4 (7%)	Program Logic Model; SWOT Framework; Quadruple Aim Framework; realistic evaluation framework	(Saunders 2022 [52]; Cullen, 2016 [71]; Santhanam 2006 [51])
Guideline	0	-	-
Tool	3 (5%)	Richard’s ecological coding procedure; self-reflective rating using simple scales of 0–10	(Reilly 2011 [49]; Rowley, 2015 [50]; Tsey 2004 [56])
*Total*	*7 (12%)*		

^1^ This grouping of ‘framework’ and ‘guideline’ are representative of the terminology noted in the related study, and this categorisation of a ‘tool’ refers to something utilised when collecting and measuring data.

### 3.3. Modes of Engagement with Aboriginal and/or Torres Strait Islander Peoples

Modes of engagement are summarized in Table 4, with the categorization determined through the data extraction stage. Just over half (53%) of the publications acknowledged the author/s as Indigenous (Aboriginal and/or Torres Strait Islander or First Nations) in their affiliations or throughout the text. A similar percentage of studies reported their mode of engagement as being through representation in a governance structure, such as in an advisory committee (39%), and through involvement in the evaluation process, such as informing the development or selection of a measurement tool (42%) (Table 4). There were often crossovers with the descriptions of engagement types. For example, when discussing representatives on steering committees as part of the governance of the project or project evaluation specifically, those representatives may also have provided guidance on the design of the evaluation. This is recorded in both related categories in Table 4 to show all of the various modes of engagement utilised as part of each evaluation.

The involvement of Aboriginal and Torres Strait Islander people in any part of the evaluation process, such as either being part of the research team or having input on the development of data collection tools was common (42%), but specific details of ways and levels of engagement was rarely described. There was limited detail on ways in which Aboriginal and Torres Strait Islander cultural input was gained across each of the categories of engagement outlined in Table 4. As an example, some studies noted that there were ‘Indigenous representatives’ on project steering committees or that ‘Aboriginal program staff’ were part of the data collection and analysis process without describing further what this looked liked in practice. Therefore, the modes of cultural engagement categorized in Table 4 are likely to represent a broad internal spectrum of degrees of engagement. Hence, it is challenging to provide strong statements of the quality of engagement from the literature due to this lack of detailed information. 

**Table 4 ijerph-20-06437-t004:** Modes of engagement with Aboriginal and/or Torres Strait Islander people in included studies ^1^.

Modes of Engagement	Number	Examples of Engagement from Included Studies
Aboriginal and/or Torres Strait Islander authors ^2^	30 (53%)	Aboriginal or Torres Strait Islanders people led or described as co-evaluators
Community representation on evaluation governance committees	22 (39%)	Inclusive of project reference groups, advisory and steering committees, community panels or locally based sub-groups that informed higher governance groups regarding the evaluation
Aboriginal and/or Torres Strait Islander people as co-evaluators or involved in an aspect of the evaluation process—design, selection or adaptation of evaluation tools/measures or analysis of findings	24 (42%)	This could include Aboriginal or Torres Strait Islanders or First Nation peoples as researchers or a collaborative development of evaluation framework, data collection tools or methodological approaches
Community-consultation-informed evaluation	6 (11%)	Examples include community feedback sessions held at points during the evaluation to provide updates and receive feedback; broad consultation with community agencies and Indigenous stakeholders; and consultation with community leaders

^1^ For studies with multiple modes of engagement, each have been recorded here. Percentages are a proportionate representation of each mode of engagement noted in studies; therefore, these will not total 100. ^2^ Where authors were noted as Indigenous within the article.

## 4. Discussion

This review shows that the number of published evaluations being undertaken in Australia informed by Aboriginal and Torres Strait Islander people and their culture has increased over time, in particular over the last three years. The majority of evaluations reported in this review have been focused on topics of health and wellbeing, with around a quarter using a resource specifically for Indigenous evaluations and more than half published by Indigenous authors showing some level of being culturally informed by Aboriginal and Torres Strait Islander peoples. 

### 4.1. Modes of Community Engagement

Modes of Aboriginal and Torres Strait Islander community engagement in evaluation processes included representation on governance groups, authorship, involvement in the evaluation processes and whole community consultation, although the latter occurred to a lesser extent. Despite all studies mentioning some level of community engagement or cultural guidance, only 24 studies provided details on Aboriginal and/or Torres Strait Islander participation in the development or conduct of the evaluation, and only 15 developed or utilised a type of Indigenous-specific evaluation guideline, tool or framework. It is worth noting, too, that ‘community engagement’ was a specific concept in the search strategy of this literature review, which implies that there is likely to be many more evaluations being conducted in the Aboriginal and Torres Strait Islander context that do not make mention of either community engagement or the incorporation of culturally specific approaches. 

### 4.2. Use of Indigenous-Specific Evaluation Resources

The limited reporting and utilization of existing Indigenous-specific evaluation resources across this literature is noteworthy, and identifying any barriers to the use and uptake would be worthy of further study. One obvious consideration is that given recent development and publication of many of the Australian evaluation resources [12,14,19,60], there is expected to be a time lag on the reporting of use of these tools, and this may be the cause of the reported encouraging upwards trend of using Indigenous-specific resources over recent years. However, as some Aboriginal and Torres Strait Islander frameworks have only become available more recently [2,60], time will tell about their uptake, useability and effectiveness in evaluation approaches in the coming years.

### 4.3. Acknowledgement of Indigenous Authorship

Studies that were published by First Nations authors accounted for just over half of included literature. Given that all studies were evaluations undertaken in an Aboriginal and Torres Strait Islander context, it could be argued that this is poor representation. However, this is likely to be an under-representation, as it represents the number of studies that specifically noted authors as Indigenous within the article or resources itself, often found in the introduction or methodology sections or acknowledgements of a paper. Better transparency and recognition of Aboriginal and Torres Strait Islander scholarship is required in published materials [76]. 

### 4.4. Reporting on Modes of Aboriginal and Torres Strait Islander Engagement

The results of this review show that there are many modes of community engagement, and this spectrum can range from consultation to collaboration and to being Indigenous-led. As with reporting on Indigenous authorship, often very little detail was provided, and the depth and quality of engagement has been difficult to assess due to poor documentation. As noted elsewhere, “*the quality of engagement matters*” [35] (p. 5), and there is a need for more detailed and standardised reporting in published literature around all Aboriginal and Torres Strait Islander modes of engagement, as well as transparency around respectful, meaningful and ethical research governance employed in studies [35,77].

As previously stated, there was a variety of modes of engagement reported in the literature; however, the actual detail and depth of engagement has been difficult to assess due to the poor documentation in evaluation papers. The lack of information means, for example, that often it cannot be determined if an evaluation is Indigenous-led or whether evaluation teams engaged with members of communities that participated in the program or service being evaluated or if Aboriginal and/or Torres Strait Islander peoples were engaged as representatives or academics without having direct experience of the programs being evaluated. Hence, it is challenging to provide a greater analysis of the engagement and difficult to present strong conclusions of the quality of engagement from the literature due to this lack of detailed information. This lack of detail also extends into the use of ambiguous terminology such as ‘program staff’. This phrase was regularly used to describe who from a service was participating in an evaluation, but often did not substantiate if service staff identified as Aboriginal and/or Torres Strait Islander peoples or if they were non-Indigenous staff members. For this review, results were only recorded where cultural identification of staff members was provided. This limited detail regarding modes of engagement is particularly problematic when some studies make broad statements such as how critical Aboriginal involvement was to the program design and evaluation. There were exceptions: for example, one Aboriginal-led evaluation described a process of commissioning external evaluators and utilizing a two-way approach between non-Aboriginal researchers and Aboriginal program staff, which included the collaborative development of an evaluation framework and program staff fulfilling cultural broker roles for the evaluators [68]. Another study clearly described the make-up and contributions of each of the project’s governance groups, as well as the respectful and transparent knowledge exchange process undertaken between researchers, where particular focus was placed on the “*process evaluation which required continual reflection and negotiation to ensure mutual understandings were reached in this cross-cultural context*” [40] (p. 152). Both evaluations highlighted the importance and responsibility of non-Indigenous evaluators and researchers using critical self-reflection skills to help ensure culturally safe practices and respectful partnerships [12]. 

### 4.5. Resourcing Culturally Safe Evaluations

On the role of funding for improving program evaluation, two perspectives emerge from the findings. The first is that evaluations need to be funded at a level that allows for early and ongoing community partnerships, input and capacity building through approaches such as co-design and participatory action research [68]. This aligns with the guiding principles for Indigenous health service program evaluations internationally [3], including Indigenous leadership or co-leadership approaches, and ensuring local Indigenous community members are part of the evaluation team. Moreover, “*… evaluation methods built on collaboration, participation, and empowerment are more effective and sustainable than those that position the researcher and members of the Indigenous community as the helper and helpee, respectively.*” [3].

The other perspective is that community-led evaluations contribute to the effective use and obtainment of funds. In one study, an Aboriginal organization who commissioned external evaluators discussed the importance of community leadership working with the outsider evaluators to ensure that, with the support of the internal project team, they were culturally aware of their role. Stated outcomes included that “*The evaluation reports were key in advocating for continued government funding, with evidence-based recommendations helping to justify the ongoing need for [staff]… This case study shows that culturally responsive evaluations can provide an avenue for Aboriginal communities to advocate for the continued funding of their programmes.*” [68]. 

### 4.6. Limitations

While not formally recognized as such, nor represented in academic journals or publications, evaluation processes such as observation, assessment and improvement cycles have always been part of Aboriginal and Torres Strait Islander peoples’ cultural practices and oral traditions [19,78]. Although a systematic process was undertaken, the broad search of the literature may not have captured all studies relevant to this review, as some papers may have not provided enough detail of their engagement approaches to meet inclusion. There is also a likely under-representation of community-led evaluation, as some internally driven evaluation work remains within organizations and is not publicly available. Limiting this review to Australian literature only misses the analysis and findings of numerous high-quality evaluations and community engagement approaches, most notably from Canadian, New Zealand and American Indigenous studies. It was felt, however, that when reviewing ways of working with Indigenous communities, the importance of historical, political, social and cultural contexts cannot be minimized; hence, the review was confined to evaluations only with Aboriginal and Torres Strait Islander communities. 

## 5. Conclusions

By reviewing the current available literature on culturally informed Aboriginal and Torres Strait Islander evaluation approaches, this scoping review provides an overview of the current state of play in the Australian context. These findings underscore the trend towards more recent development of Indigenous-specific evaluation resources and a variety of types of engagement with Aboriginal and Torres Strait Islander peoples. However, with literature providing limited details on the spectrum of culturally informing engagement approaches, we are unable to critically review the quality and effectiveness of evaluations based on Aboriginal and Torres Strait Islander leadership, engagement or valuing of their culture. 

Results also show that there is only a small, yet recently growing, number of Indigenous-specific evaluation frameworks, tools or guidelines developed, as well as a dearth of published examples of the uptake and utilization of these existing evaluation resources in Australia’s Aboriginal and Torres Strait Islander context. The findings suggest that a gap remains in the conduct of evaluations that are informed by First Nation paradigms. While common research methodologies such as co-design and participatory action research were regularly reported in detail, there was a limited number of studies that either (a) adopted an evaluation resource that allowed for the centralization of Aboriginal and Torres Strat Islander ways of knowing, being and doing or (b) provided adequate details on community representation and engagement processes that were used to inform the evaluation approach and, in so doing, how Aboriginal and Torres Strait Islander perspectives were foregrounded in the evaluation approach.

One intention of this review is to inform and contribute to further improvements to program evaluation approaches in the Aboriginal and Torres Strait Islander context through stronger adoption of Indigenous-led and informed evaluation methods and guidelines that include respectful partnerships and engagement approaches, the development and application of culturally informed evaluation resources and improved reporting processes to share learnings about culturally responsive evaluation processes. This will contribute to strengthening effective and safe evaluation approaches for the development of reliable and relevant evidence to drive program and policy improvements that benefit communities. 

Our recommendations include further development of Indigenous evaluation tools and guidance, the increased undertaking and publishing of culturally informed evaluations that utilise these resources, as well as the development of standardized reporting criteria that provide greater transparency and detail around cultural engagement methods and approaches. This future focus and research will contribute to the overall improvement of culturally safe and responsive evaluation practices that are culturally relevant to Aboriginal and Torres Strait Islander communities. 

## Figures and Tables

**Figure 1 ijerph-20-06437-f001:**
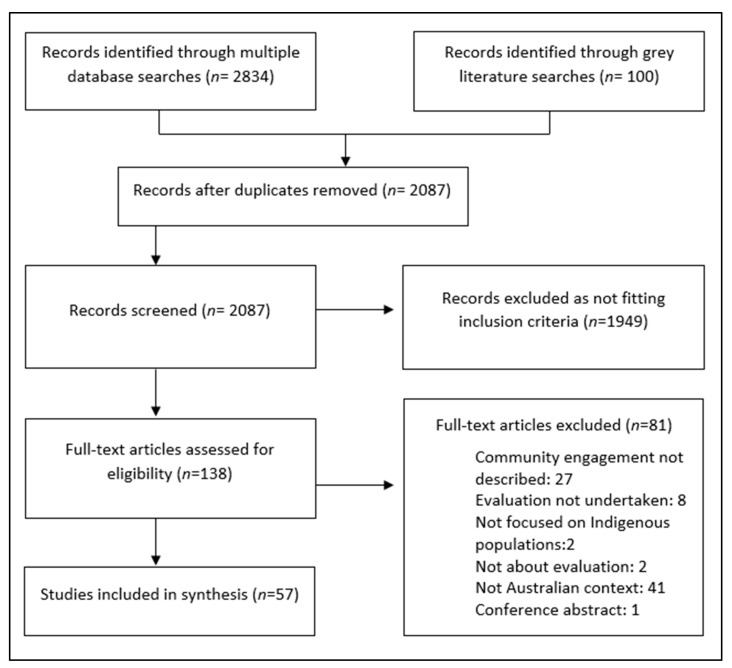
PRISMA flow diagram of the search strategy and study inclusion.

**Figure 2 ijerph-20-06437-f002:**
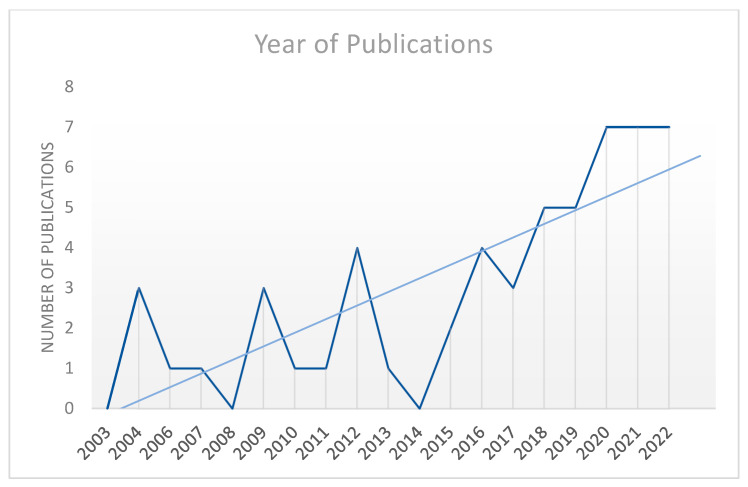
Number of articles covered in this review per year.

## Data Availability

All relevant data within the manuscript are available from corresponding author upon reasonable request.

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
