# Peer review of "Culturally Informed Australian Aboriginal and Torres Strait Islander Evaluations: A Scoping Review"

_ijerph, 2023, doi:10.3390/ijerph20146437_

Round 1

Reviewer 1 Report

This is a well-written, meaningful, and timely paper. My only concern is that the authors often conflate research methods/methodology with evaluation methods/methodology. While there are definite similarities it would be helpful to maintain boundaries where relevant.

Author Response

Reviewer 1

Comment

This is a well-written, meaningful, and timely paper. My only concern is that the authors often conflate research methods/methodology with evaluation methods/methodology. While there are definite similarities it would be helpful to maintain boundaries where relevant.

Response

Thank you for your consideration and feedback. In response we have reviewed the manuscript and made amendments where it appears there is conflation between description of research or evaluation methodologies, such as clarification provided in lines 124 and 388. In summarising and analysing the diversity of the included literature, it was necessary to talk about both research and evaluation methodologies together. For example when discussing the need for adequate resourcing in line 331, this relates to building strong relationships and in both research and evaluation methodological approaches.

Reviewer 2 Report

This is a very interesting article and I thoroughly commend the authorship team. Overall, it is extremely well written. The search strategy is sound and the search methodology and results are clearly presented. Conclusions drawn from this scoping review accurately reflect the findings from the literature. This research presents very interesting points that should be considered for future research involving evaluation of programs or service delivery specifically intended for Aboriginal and Torres Strait Islander people, particularly regarding sufficient funding requirements and standardised reporting on the involvement and engagement of Aboriginal and Torres Strait Islander people. 

My only suggestion is, is there a reference for the following statement? "While not formally recognized as such, nor represented in academic journals, evaluation processes have been part of Aboriginal and Torres Strait Islander peoples’ cultural practices and oral traditions for millennia."

Author Response

Reviewer 2

Comment

This is a very interesting article and I thoroughly commend the authorship team. Overall, it is extremely well written. The search strategy is sound and the search methodology and results are clearly presented. Conclusions drawn from this scoping review accurately reflect the findings from the literature. This research presents very interesting points that should be considered for future research involving evaluation of programs or service delivery specifically intended for Aboriginal and Torres Strait Islander people, particularly regarding sufficient funding requirements and standardised reporting on the involvement and engagement of Aboriginal and Torres Strait Islander people. 

My only suggestion is, is there a reference for the following statement? "While not formally recognized as such, nor represented in academic journals, evaluation processes have been part of Aboriginal and Torres Strait Islander peoples’ cultural practices and oral traditions for millennia."

Response

Thank you for taking the time to review our paper and for your generous feedback, it is much appreciated. As suggested, I have now added references to this slightly amended statement giving recognition to Aboriginal and Torres Strait Islander peoples as the original evaluators.

Reviewer 3 Report

Thank you for the opportunity to be part of the editorial process of this work and I hope my comments may be useful. I command the authors for their efforts with potentially relevant implications for policy and practice.

I would ask the authors to provide a clear definition of culturally informed evaluations and what these evaluations refer to.

I also miss a better focus on the local context and how this work advances the field - why this study is relevant and what novel contributions brings. What we know about this context and how/why relevant to truly advance the field? What incremental conceptual and methodological contributions? Thus, the introduction needs to provide a strong rationale for this study, while also convincing the reader of its novelty, relevance and uniqueness.

As to methods, it remains unclear what selection criteria have been used, why this method for data collection and analyses and which and why specific analyses were performed.  What types of articles, why this time frame, how selected and analysed, by whom, what interrater agreement etc.

Author Response

Reviewer 3

Comment 1

Thank you for the opportunity to be part of the editorial process of this work and I hope my comments may be useful. I command the authors for their efforts with potentially relevant implications for policy and practice.

I would ask the authors to provide a clear definition of culturally informed evaluations and what these evaluations refer to.

Response 1

Thank you for taking the time to review this paper and for your suggestions. We have expanded the Materials and Methods section, line 165, to more clearly describe what is meant by culturally informed evaluation in this paper.  

Comment 2

I also miss a better focus on the local context and how this work advances the field - why this study is relevant and what novel contributions brings. What we know about this context and how/why relevant to truly advance the field? What incremental conceptual and methodological contributions? Thus, the introduction needs to provide a strong rationale for this study, while also convincing the reader of its novelty, relevance and uniqueness.

Response 2

The novelty of this review is in providing a synthesis and analysis of the growing number of Australian Indigenous evaluations, and to examine what relevant guidelines and tools exist and how they are being applied. We presented the rationale in the introductory statement (from line 126), highlighting the value of this review in contributing to the evidence and guidance of programs and research affecting Aboriginal and Torres Strait Islander peoples of Australia. The relevance of this work links to a shift internationally towards more Indigenous-led evaluations and development and use of Indigenous evaluation resources and guiding principles (noted in line 101). This paper specifically focuses on the Australian current state of play to add new insight into how Aboriginal and Torres Strait Islander knowledges, perspectives and culture can inform evaluations, how First Nation Australian’s are engaged in the development and implementation of evaluations, as well as the current gap of culturally informed evaluations as shown through the analysis of the literature. The three recommendations of this review, as stated in the abstract and further in the conclusions copied below, provide readers with actionable focus areas for future research.  

“Our recommendations include further development of Indigenous evaluation tools and guidance, the increased undertaking and publishing of culturally informed evaluations that utilize these resources, as well as the development of standardized reporting criteria that provides greater transparency and detail around cultural engagement methods and approaches.”

Comment 3

As to methods, it remains unclear what selection criteria have been used, why this method for data collection and analyses and which and why specific analyses were performed.  What types of articles, why this time frame, how selected and analysed, by whom, what interrater agreement etc.

Response 3

We note that Reviewer 2 stated that “The search strategy is sound and the search methodology and results are clearly presented.” and hope to provide some clarity to Reviewer 3’s comments here. The method followed for this review was the PRISMA extension for scoping reviews, noted in line 148. Inclusion criteria are described from line 164, and exclusion criteria from line 169. The justification for the timeframe for the search strategy being two decades was informed by the number of results from an exploratory pilot scope of the available literature, as described from line 150, of both the peer-reviewed and grey literature using databases and an online citation-based literature mapping tool. Included literature were peer-reviewed and online grey literature, limited to PDF files, that were written in English. There was also exclusion of articles describing evaluations that had not yet been undertaken, such as study protocols, to allow for equal comparisons and analysis only of conducted evaluation approaches and results, see line 175. We have added mention of the approach to inter-rater agreements in resolving conflicts, and feel the level of detail on the two-step process for screening the literature is sufficiently described from line 162. The three authors involved in data extraction for results analysis are noted in line 181. Based on your comments, some additional details have also been added to this section of the method description to ensure greater clarity.